# New continuous total ozone, UV, VIS and PAR measurements at Marambio 64°S, Antarctica

Kaisa Lakkala[1,2], Margit Aun[1,3], Ricardo Sanchez[4], Germar Bernhard[5], Eija Asmi[1,4], Outi Meinander[1], Fernando Nollas[4], Gregor Hülsen[6], Tomi Karppinen[2], Veijo Aaltonen[1], Antti Arola[1], and Gerrit de Leeuw[1]

[1]Finnish Meteorological Institute, Climate Research Programme
[2]Finnish Meteorological Institute, Space and Earth Observation Centre
[3]University of Tartu, Estonia
[4]Servicio Meteorológico Nacional, Argentina
[5]Biospherical Instruments Inc., San Diego, USA
[6]Physikalisch-Meteorologisches Observatorium Davos, World Radiation Center, Switzerland

*Correspondence to:* Kaisa Lakkala (kaisa.lakkala@fmi.fi)

**Abstract.** A GUV multifilter radiometer was set up at Marambio, 64°S 56°W, Antarctica, in 2017. The instrument measures continuously ultraviolet (UV) radiation, visible (VIS) radiation and photosynthetically active radiation (PAR). The measurements are designed for providing high quality long-term time series which can be used to assess the impact of global climate change in the Antarctic region. The quality assurance includes regular absolute calibrations and solar comparisons performed

at the Marambio and at Sodankylä, Finland. The measurements continue observations at Marambio that were performed with NILU-UV radiometers between 2000 and 2010 as part of the Antarctic NILU-UV network. These measurements are ideally suited for assessing the effects of the ongoing stratospheric ozone recovery on the ecosystem as the data products include information on radiation at various wavelengths ranging from UV to VIS so that changes on biologically effective radiation due to ozone can be separated from those due to other factors. Data products include total ozone, photosynthetically active radiation

(PAR), visible (VIS) radiation at 555 nm, UV index, UV irradiance at 5 channels, UVB and UVA dose rate/daily dose, and biologically weighted UV dose rate/ daily dose, calculated with 10 different action spectra. The data from the last five days and the daily maximum UV index time series are plotted and updated daily on the web page fmiarc.fmi.fi/sub_sites/GUVant/. The first two years of UV measurements were very different in terms of the results: For October, November and December the monthly average of daily maximum UVB dose rates were clearly higher in 2018 than in 2017. The largest difference was

observed in October, when the average of daily maximum UVB dose rates was 76 $\mu$W cm$^{-2}$ and 102 $\mu$W cm$^{-2}$ in 2017 and 2018, respectively. Monthly averages of the three months were similar in 2018, while in 2017 the monthly average of October was lower than those of November and December. VIS and PAR time series show that daily maxima in 2018–2019 exceed those in 2017–2018 during late spring and summer (mid-November – January). The studied dataset, including daily maximum irradiances at five UV channels and one VIS channel, daily maximum UVB/UVA and PAR dose rates, noon UVB/UVA and PAR

dose rates, noon total column ozone and UVB/UVA daily doses, is freely accessible at http://doi.org/10.5281/zenodo.3688700 (Lakkala et al., 2019).

# 1 Introduction

The Antarctic area suffers from the well-known springtime ozone hole since the late 1970s (e.g., Farman et al., 1985; WMO, 1990). Increase of harmful UV radiation reaching the surface was expected, as the stratospheric ozone layer protects the Earth from the most dangerous ultraviolet (UV) wavelengths. The international Montreal Protocol was signed in 1987 to restrict the use of ozone depleting substances and thus protect the whole ecosystem from excessive UV levels (e.g., Barnes et al. (2019); WMO (2018b)). The shortest UV wavelengths are known to cause skin cancer, sunburn and cataract, and they can also harm plants, animals and materials (e.g., EEAP (2019)).

The Spanish-Finnish-Argentinean Antarctic NILU-UV network was established in 1999/2000 to monitor UV radiation, photosynthetically active radiation (PAR) and total ozone in the Antarctic area and to serve the multidisciplinary UV research community (Lakkala et al., 2008). The NILU-UV multifilter radiometer measurements stopped due to degradation of the instrument at Marambio and other stations in 2013, but they provide reference time series for the severe Antarctic ozone hole period. Between years 2000–2010 the UV index at Marambio reached a maximum of 12 when the station was inside the ozone hole (Lakkala et al., 2018b).

The latest Scientific Assessment of Ozone Depletion (WMO, 2018b) suggests that the Montreal Protocol has been successfully adapted and the Antarctic ozone hole has started to recover even though it continues to occur each year. To detect the recovery and its effects on UV radiation and the ecosystem, the Finnish Meteorological Institute (FMI) and the Servicio Meteorológico Nacional Argentina (SMN) started UV and total ozone measurements with multifilter GUV radiometers at Marambio in 2017. As the stratospheric ozone depletion has influenced both stratospheric and surface climate, e.g., cooling of Antarctic stratosphere and southward shift of mid-latitude rain, and the Southern Ocean temperature and circulation, the recovery is expected to have the opposite effects (WMO, 2018a). However the influence of the ozone recovery on climate and ocean cannot easily be predicted as the influence will depend on the evolution of green house gas concentrations in the atmosphere which is the key driver of future southern hemisphere climate (WMO, 2018b).

The total ozone, UV radiation, visible (VIS) radiation and PAR measurements of the GUV multifilter radiometer can be used in research assessing the effects of the ongoing climate change. For example, cloud optical depth can be retrieved using measured irradiances (Dahlback, 1996) and, together with the irradiance of the visible channel, cloud optical depth can give information on changes in cloudiness. PAR measurements are directly applicable in studies of the effects of climate change on the photosynthesis of plants, algae and bacteria. Both UV and PAR affect micro-organisms living in Antarctic ice and the Southern Ocean (Deppeler and Davidson, 2017; Häder et al., 2014). Changes in the amount of aerosols and pollution as well as changes in sea ice extension or ground albedo are also reflected in both UV and VIS radiation time series (Fountoulakis et al., 2014; Wild, 2009).

In 2013, the FMI installed Solar Light 501A radiometers at Marambio to monitor both incoming and outgoing UV radiation. The measurements are used to determine local albedo and detect changes in it. They can be used to link the NILU-UV measurement time series to the new GUV measurements as they monitor the erythemally weighted UV irradiance, which was one product of the NILU-UV radiometer measurements and is now one of the GUV radiometer products. The Solar Light 501A

data collection system consists of one upward and one downward radiometer, which measure the irradiance weighted with the action spectrum for UV induced erythema (McKinlay and Diffey, 1987), which in turn also has a contribution from the UVA. The radiometer pair is selected to represent as similar spectral and cosine responses as possible, as demonstrated in Fig. 1 of Meinander et al. (2008). The system measures in one minute intervals and saves data automatically.

5     In this paper, the new UV, VIS, PAR and total ozone measurements at Marambio are described and the quality assurance procedures are discussed. Time series of the first two years of measurements are shown.

## 2   GUV multifilter radiometer measurements at Marambio

GUV multifilter radiometer was installed at Marambio, Antarctica, in collaboration between the SMN and the FMI, in March 2017. The SMN is responsible for the operation and quality control of the measurements at the station, while the FMI is 10  responsible for the quality assurance, data dissemination and data storage.

### 2.1   Marambio station

Marambio station in Antarctica is located on the highest part of the Seymour/Marambio ice-free Island, surrounded by Weddell sea, at $64^{\circ}$ $14'$ S; $56^{\circ}$ $37'$ W, on the north-east side of the Antarctic Peninsula. The altitude of the station is 198 m above the sea level. During the last decades the station is well-known for studies related to the ozone hole phenomena because it is located at the edge of the polar vortex. The temperatures at the site are between $+10°C$ during the summer, and $-30°C$ during the winter, though strong winds can lower the apparent temperature down to $-60°C$. During most part of the year, the soil is frozen and covered with snow. The Weddell Sea to the east of the Peninsula is frozen year round. During the winter months the ice may extend to latitudes of around $60°S$, also covering the coast of Marambio Island. The prevailing wind directions are from the southwest and the northwest. The wind speed can reach values close to 100 km/h. Heavy cloudiness and fog are common 20  during the summer months, and heavy winds in the winter can blow the snow away yielding to lifting of dust into the air. The station is part of the Global Atmospheric Watch (GAW) program of the World Meteorological Organization (WMO). More information on the site can be found from earlier publications (e.g., Karhu et al., 2003; Asmi et al., 2018).

    UV measurements in Marambio have been conducted with a NILU-UV multichannel radiometer between 2000 and 2013 (Lakkala et al., 2018b), and before that, measurements of erythemal UV radiation were carried out by a biometer Solar Ligth 25  501 installed in 1996 by SMN. The instrument was calibrated by personnel from the World Radiation Center (WRC) in 2006 and 2010. In 2013 it was replaced by a similar instrument belonging to the FMI, when also a radiometer was installed to measure outgoing solar UV radiation (Meinander et al., 2014). Details about the calibration of the new radiometers can be found in Meinander et al. (2008).

    Synoptic observations are performed hourly at the Marambio station and additional weather data is obtained from an au-30  tomatic station. In addition to the synoptic visual cloud observations, cloud coverage and height in Marambio are monitored since the year 2016 with a ceilometer model CL51 (Vaisala), with 30 second resolution.

Marambio also reports to the World Ozone and UV Data Center (WOUDC) and has additionally programs to monitor greenhouse gas concentrations and various aerosol parameters. The number concentration of aerosol particles larger than 10 nm in radius is measured with a Condensation Particle Counter (CPC) model 3772 (TSI Inc.) installed as a part of the particle size distribution measurement system. Aerosol optical properties at the surface are measured with online instrumentation: scattering with an Aurora 3000 nephelometer and absorption with a Multi-Angle Absorption photometer (MAAP) model 5012 (Asmi et al., 2018). Aerosol chemistry is measured offline from the collected weekly filter samples (see e.g., Asmi et al., 2018). Aerosol optical depth in the atmospheric column is measured with a PFR sun photometer which is part of the GAW-PFR network (Tomasi et al., 2015).

## 2.2 GUV radiometer

The radiometer installed in Marambio is a GUV radiometer, model GUV-511, manufactured by Biospherical Instrument Inc. (BSI), USA, which monitors UV, visible and PAR radiation. It includes six channels which central wavelengths are at 305, 313, 320, 340, 380 and 555 nm, and the full width at half maximum (FWHM) is around 10 nm. In addition there is a seventh channel which measures PAR in the 400–700 nm wavelength region. Using the combination of different channels offers the possibility to retrieve cloud optical thickness and total ozone column. The instrument is environmentally sealed and temperature-stabilized at 40°C. It has a Teflon-covered quartz cosine collector. The instrument specifications, as provided by the manufacturer, are shown in Table 1. Details of the GUV radiometer and its performance are described in Bernhard et al. (2005).

**Table 1.** GUV model 2511 irradiance array specifications.

| | |
|---|---|
| Filter type | Custom low-fluorescence interference |
| Cosine collector | Teflon -covered quartz |
| Collector area | 2.1 cm diameter |
| Out-of-band rejection | $1 \times 10^{-6}$ |
| Angular response | 0-5% from $0°$ to $70°$; $\pm 10\%$ from $71°$ to $85°$ |
| Typical Saturation | $10^5$ $\mu$Wcm$^{-2}$ nm$^{-1}$ |
| Noise equivalent irradiance | $10^{-11}$ Wcm$^{-2}$ nm$^{-1}$ |
| Temperature coefficient of the dark Signal | Less than $\pm 3$ $\mu$volts/°C |
| Response Temperature Coefficient | Less than $\pm 0.15$ %/°C |

## 2.3 Data retrieval and products

The sampling rate of the GUV radiometer can be set to 1-20 Hz and one minute averages are automatically recorded using LOGGER data-acquisition software. The software converts raw data into irradiances, which are saved as ascii and Microsoft Access Database formats. The daily data is automatically transferred to an ftp server in the SMN, from which the data is transferred and stored into the FMI's data base. Measurements from the last five days and the daily maximum UV index time

series are plotted and updated daily on the web page fmiarc.fmi.fi/sub_sites/GUVant/. Examples of these web plots including UV index, UVB, UVA and PAR radiation are shown in Fig. 1. In addition to routine checks of data transfer, the housekeeping includes cleaning of the diffusers at least once a week, or more frequently if needed, and checking of the leveling once a month.

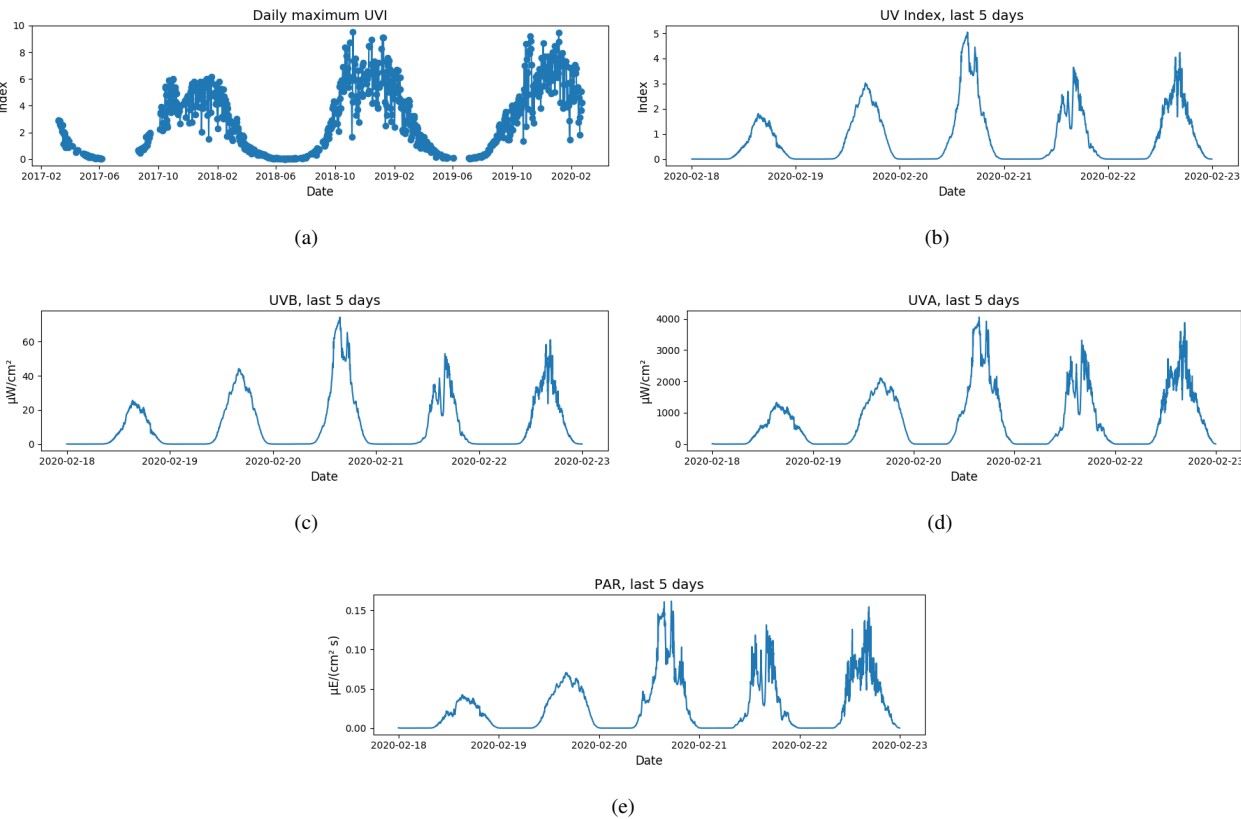

**Figure 1.** Near real time plots updated daily: A) Daily maximum UV index time series, b) UV index, c) UVB irradiance, d) UVA irradiance and e) PAR. Data in plots b–e are one minute averages.

5     Near real time data processing includes calculation of data products based on UV, VIS and PAR measurements. The main idea is that each UV product is calculated using a linear combination of calibrated irradiances measured at the five UV channels. The coefficients are determined from solar comparisons between the GUV radiometer and a spectroradiometer following the method of Dahlback (1996) and Bernhard et al. (2008) as explained in Section 3.1. Bernhard et al. (2005) showed that GUV multifilter UV data agree within 5% from data from a well calibrated spectroradiometer for solar zenith angles (SZAs) smaller
10   than 80°. In addition to UV products, VIS radiation and PAR are measured using the 555 nm and 400–700 nm channels, respectively. Total ozone is calculated using lookup tables and the ratio of irradiance measured at 305 nm (a wavelength strongly attenuated by ozone) and 340 nm (a wavelength weakly attenuated by ozone). The complete list of products including

several biologically weighted UV dose rates and doses and their references can be found in Table 4.4. of Bernhard et al. (2008). Table 2 summarizes the products calculated from the GUV measurements at Marambio.

**Table 2.** Data products of the GUV radiometer at Marambio.

| Product | Wavelengths [nm] / Biological effect |
| --- | --- |
| UV Irradiance | 305, 313, 320, 340, 380 |
| UVB and UVA dose rate and daily dose | 290–315, 290–320, 315–360, 320–360, 360–400, 315–400, 320–400 |
| Visible irradiance | 555 |
| Photosynthetically active radiation (PAR) | 400-700 |
| UV Index | |
| Total ozone | |
| Biologically weighted UV dose rate and daily dose | Erythema, DNA damage, skin cancer in mice, skin cancer in mice corrected for human skin, generalized plant damage, plant growth, damage to anchovy, inhibition of phytoplankton carbon fixation, Inhibition of phytoplankton photosynthesis of phaeodactylum and prorocentrum, inhibition of photosynthesis in Antarctic phytoplankton |

## 3 Quality assurance of the measurements

The quality assurance of the GUV measurements is based on regular absolute calibrations and solar comparisons against high quality spectroradiometers. The FMI has purchased two GUV radiometers in order to avoid gaps in measurement time series. Those two instruments rotate so that one is measuring at Marambio, while the other one is calibrated and compared against reference instruments. Solar comparison is performed each time when the instruments are switched, to detect possible drift or jump in the calibration scale.

Solar comparisons against spectroradiometers are performed at Sodankylä 67°N, in Finland. As this site is located at a high latitude, similar to Marambio, the atmospheric path of UV radiation reaching the surface is similar at both locations: long path in the atmosphere due to large SZAs, which means more ozone absorption and scattering than at smaller SZAs. Sodankylä can be classified as an Arctic site from the point of view of stratospheric meteorology. This means that, as Marambio, the station can be located inside, outside or at the edge of the stratospheric Arctic polar vortex, and severe spring time stratospheric ozone loss can occur.

Solar comparisons make it possible to use comparable irradiance scales in both Marambio and Sodankylä. For example, by comparing UV levels at the Arctic and Antarctic location, the effect of ozone loss events on UV radiation can be studied.

## 3.1 Absolute calibration

Both GUV-2511 radiometers (serial numbers #162 and #163) were vicariously calibrated at BSI in April 2016 against measurements of a SUV-100 spectroradiometer that is part of the NSF UV Monitoring Network (Booth et al., 1994). GUV and SUV instruments were installed side by side on the roof platform of BSI in San Diego, California. The SUV-100 measures spectra of global irradiance between 290 and 600 nm with a spectral resolution of 1 nm. The GUV radiometer with serial number 163 was recalibrated at BSI in April/May 2019 using the same approach.

The calibration method is identical to the method described in Section 4.3 of Bernhard et al. (2008). The procedure is based on the work by Dahlback (1996). In brief, uncalibrated, dark signal corrected measurements of each channel of the GUV radiometers are regressed against cosine error corrected measurements of the SUV-100 spectroradiometer. Measurements of the SUV-100 are weighted with the spectral response functions of the GUV prior to performing the regression. The procedure results in a calibration factor $k_i$ for each channel $i$ of the GUV. Calibration factor established for GUV #163 in 2016 and 2019 agree to within $\pm 2\%$ with the exception of the channel at 305 nm, which drifted by $\pm 5\%$.

Calibrated measurements are finally calculated by dividing the GUV's raw data with these calibration factors. Measurements of the GUV's PAR channel are calibrated slightly differently (Bernhard et al., 2008).

Because the SUV-100 data were weighted with the GUV's response functions, calibrated measurements of the GUV are "response-weighted" irradiances (Seckmeyer et al., 2010). The conversion from response-weighted irradiance to useful data products $D$, such as spectral irradiance for a given wavelength, erythemal irradiance, or the UV index, is performed with the method suggested by Dahlback (1996). In brief, $D$ is approximated by a linear combination of the dark signal corrected signals of the GUVs' UV channels $V_i$:

$$D = \sum_{i=1}^{5} a_i V_i, \tag{1}$$

where the coefficients $a_i$ depend on the calibration factors $k_i$ and the action spectrum of the biological effect of interests (e.g., erythemal response). The coefficients are determined by solving a system of linear equations as described by Bernhard et al. (2008), taking into account the conditions at the deployment site (e.g., range of total ozone and surface albedo).

The uncertainty of GUV data products is composed of (i) the uncertainty of SUV-100 measurements, (ii) the uncertainty of the transfer of the calibration from the SUV-100 to the GUV, (iii) the uncertainty of the conversion from response-weighted irradiance to data products D, and (iv) the drift of GUV calibrations. The uncertainty of SUV-100 measurements has been assessed by Bernhard et al. (2004). The expanded uncertainty (coverage factor k = 2, corresponding to a level of confidence of approximately 95 %) of the UV index and DNA-damaging irradiance varies between 5.8 and 6.4 %. The upper limit of errors in the SUV-to-GUV calibration transfer was estimated to $\pm 2$ % from the reproducibility of the vicarious calibration method. The uncertainty in calculating the UV index from response-weighted irradiance using Eq. (1) was assessed by Dahlback (1996). For SZAs between 0 and 80° and total ozone between 200 and 500 DU, the approximations implied in using Eq. (1) agreed to within $\pm 5$ % with exact radiative transfer calculations. However, larger errors were found for total ozone columns smaller than 200 DU or for SZAs larger than 80°(when absolute values of the UV Index are small). Bernhard et al. (2005) compared measurements

of UV-B irradiance, UV-A irradiance, the UV index, and DNA-damaging performed at the South Pole with a SUV-100 and a GUV radiometer, which was calibrated with the method described in this paper. For SZAs smaller than 80°, measurements by the two instruments agreed on average to within ±2.5 %, and the standard deviation of the ratio of GUV/SUV data was smaller than 4.0 % for the four data products. The magnitude of these variations is in good agreement with the theoretical calculations by Dahlback (1996). Lastly, the uncertainty attributed to drifts was calculated from the observed change of 5 % in the responsivity of the GUV's 305 nm channel. The four uncertainty components were combined in quadrature and multiplied with a coverage factor of k = 2. For SZAs smaller than 80°, the expanded (k = 2) uncertainty is 9 % for UV-B irradiance, the UV index, and DNA-damaging irradiance.

Total ozone column is calculated from GUV measurements with lookup tables, which relate total column ozone to SZA and the ratio of GUV measurements at 305 and 340 nm. The retrieval algorithm is similar to the method described by Stamnes et al. (1991). Lookup tables are calculated with the radiative transfer model UVSPEC/libRadtran (Mayer and Kylling, 2005) and resulting model spectra are weighted with the GUV response functions at 305 and 340 nm. On average, GUV total ozone data agree with OMI data to within ±5% for SZA smaller than 75°. At larger SZA, differences become greater due to the dependence of the retrieval on the vertical distribution of ozone in the atmosphere. Also noise in the GUV 305 nm channel affects the ability to calculate total ozone with good precision at large SZAs. A systematic comparison of GUV and TOMS total ozone measurements at several sites was performed by Bernhard et al. (2005). For SZAs < 80°, the bias between GUV and TOMS measurements is less than 5 %.

## 3.2 Comparisons against spectroradiometers at Sodankylä

The UV radiation and ozone measurements of the GUV radiometers were compared against high quality spectroradiometer measurements during each Northern Hemisphere summer since 2016 (Table 3). The comparisons were performed at the measurement platform next to the sounding station of the FMI Arctic Space Centre in Sodankylä. The FMI has two Brewer spectroradiometers, which routinely measure UV spectra about every half hour (Mäkelä et al., 2016). The measurements are cosine and temperature error corrected (Lakkala et al., 2018a) and the irradiance scale is traceable via the National Standard Laboratory MIKES, Aalto University, Finland, to the scale maintained by the National Research and Testing Institute (SP) (Heikkilä et al., 2016; Lakkala et al., 2008). As the measurement of a Brewer UV spectrum takes around 3 minutes while the GUV radiometer records irradiances averaged over one minute, the three minutes averages of GUV measurements were used in the comparison of UV measurements. UV indices and UVB dose rates calculated from Brewer #37 spectra and from GUV irradiances were compared and results are shown as function of SZA in Figs. 2 – 3. As the Brewer #37 measures only until wavelengths up to 325 nm, only the comparison of the UVB part of the UV spectrum is possible. For the calculation of Brewer UV indices, as explained in Mäkelä et al. (2016), a predefined UVA spectrum is scaled to the last wavelength to complement the measurements and thus take into account the whole effective wavelength range. The erythemal weighting function, which is used in the calculation of UV indices, approaches zero towards longer UVA wavelengths and the uncertainty caused by this UVA extension is only of the orfer of $10^{-1}$ (Mäkelä et al., 2016).

The first comparison was made in August 2016, shortly after the radiometers were purchased and before the first one was sent to Marambio. For clear sky and SZAs smaller than 60°, UV indices measured by both GUV radiometers agreed within 1% of each other and the Brewer spectroradiometer. The differences for UVB dose rates ranged from 3 to 5 %. After the comparison, the GUV radiometer #163 was sent to Antarctica and continuous measurements started there in March 2017.

The GUV radiometer #162 stayed in Sodankylä and was compared with the Brewer UV indices during the next years, in June 2017 and July 2018. The sky was clear from clouds during the comparisons. Both UV indices and UVB dose rates agreed within 3% for SZAs smaller than 60° in 2017. In 2018, the differences in UV indices and UVB dose rates ranged from 2 to 6 % and from 0 to 3 %, correspondingly, depending on the SZAs. The SZA dependency was due to the non-corrected angular response of the GUV radiometer. Non-perfect leveling might have caused small azimuth dependency in 2018.

The portable UV world reference spectroradiometer QASUME from the World Calibration Center for UV (WCC-UV) at the Physikalisch-Meteorologisches Observatorium Davos, World Radiation Center (PMOD/WRC) visited Sodankylä for a site audit in June 2018. QASUME is a double monochromator spectroradiometer, whose solar UV irradiance measurements are traceable to the primary spectral irradiance standard of the Physikalisch-Technische Bundesanstalt (PTB), Germany, (Gröbner and Sperfeld, 2005). The measurements are temperature stabilized and the expanded relative uncertainty (coverage factor

k=2) of solar UV irradiance measurements is 3.1% for SZAs smaller than 75° (Hülsen et al., 2016). The GUV #162 was measuring on the site during the audit and the ratios of UV indices derived from it's measurements and QASUME's spectra were calculated. The results are shown in Fig. 2. The weather was unstable during the whole week when the comparisons were made, with changing cloudiness and rain. The day with the thinnest clouds was chosen and spectra measured under rapidly changing cloudiness were excluded from the analysis. The results show that the GUV radiometer UV indices were within

±5% from the QASUME UV indices for SZA smaller than 60°. Although measurements performed during rapidly changing cloudiness were excluded from the analysis, the influence of changing cloudiness during the Brewer scan can be indentified as the outlier in Fig. 2, at around SZA 56° on 5 June 2018.

**Table 3.** Solar comparisons at Sodankylä, Finland.

| Date | GUV radiometer | Spectroradiometers |
|------|----------------|--------------------|
| August 2016 | GUV162, GUV163 | Brewer037, Brewer214 |
| June 2017 | GUV162 | Brewer037, Brewer214 |
| June 2018 | GUV162 | Brewer037, Brewer214, QASUME Bentham |
| July 2018 | GUV162 | Brewer037, Brewer214 |

### 3.2.1   Total ozone column

The total ozone column measured by the GUV radiometer was compared to Brewer measurements during the same days than

the UV comparisons were performed in Sodankylä. The Brewer #214 was used as a reference instrument as it has proven good performance during international calibrations and comparison campaigns. The Brewer participated in the international

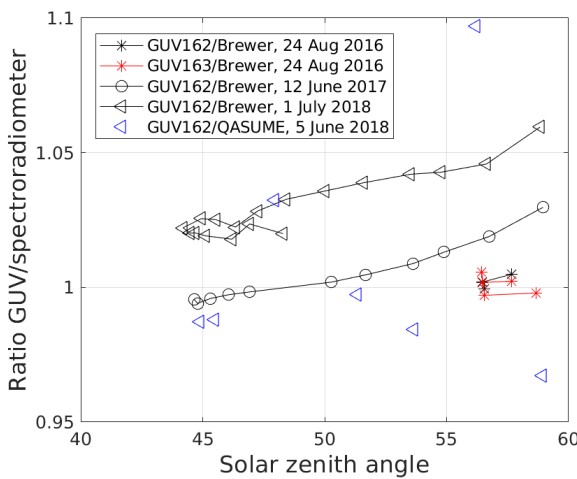

**Figure 2.** Ratio of the GUVs' and spectroradiometers' UV index at Sodankylä.

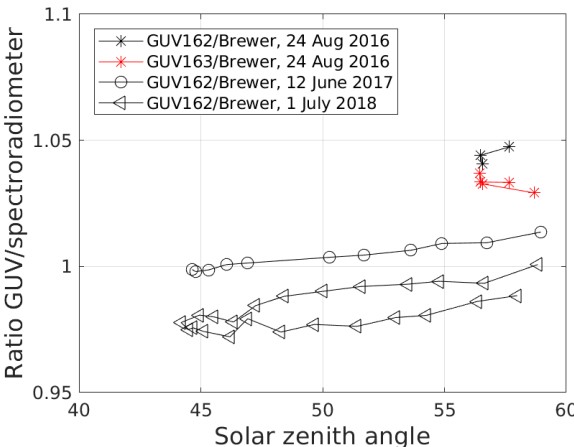

**Figure 3.** Ratio of the GUVs' and Brewer spectroradiometer's UVB dose rates at Sodankylä.

comparison campaign held in Huelva, Spain in May 2017. The results showed that the instrument agreed to the reference within 1% (Redondas et al., 2019) after proper calibration. The calibration is maintained by regular, every second year, visits of the International Ozone Service (IOS) to perform the maintenance and calibration at the measurement site.

The Fig. 4 shows the results of the comparisons in 2016, 2017 and 2018. For the Brewer radiometer, only direct sun mea-

5    surements with a standard deviation less than 2.5 DU were included in the study. The nearest, which was within one minute, GUV radiometer measurement was chosen for the study. The total ozone column measurements of the GUV and the Brewer agreed to within ± 2% for SZAs smaller than 60°.

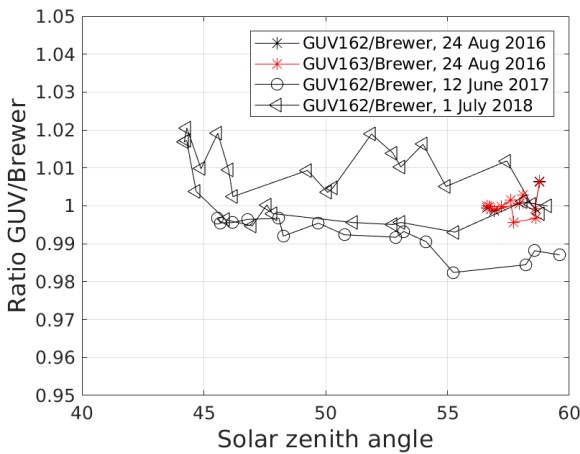

**Figure 4.** Ratio of the GUVs' and Brewer total ozone column at Sodankylä.

### 3.3 Solar comparison at Marambio

At Marambio, the first exchange of the GUV radiometers occurred in November 2018. The GUV #162 was set up next to the GUV #163, which had measured continuously for nearly two years at Marambio. The two instruments measured simultaneously for two weeks before the GUV #163 was packed and shipped for recalibration. The results of the comparison show a difference of 4 – 6% for SZAs smaller than 60° (Fig. 5). Such differences were expected, as it is common for filter radiometers that the wearing-in of Teflon diffuser introduces small changes in the response of channels during the first years of measurements. The drift was considered to be within the uncertainties of the measurements and the time series was not corrected for.

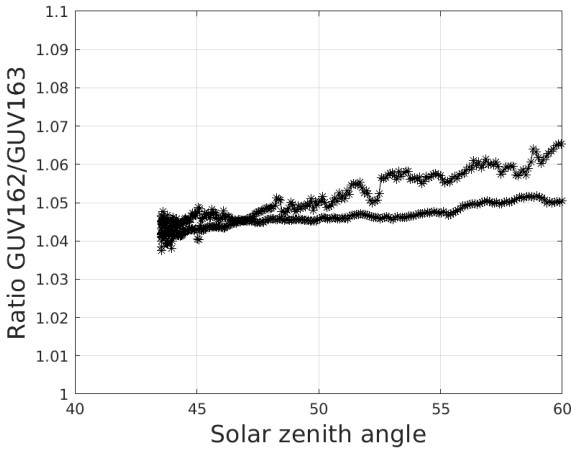

**Figure 5.** Ratio of the GUVs' UV index at Marambio on 25 November 2018.

## 4 First two years of measurements

The erythemally weighted UV, maximum UV index and daily average total ozone time series were discussed in Aun et al. (2019) and the results were compared to measurements from other Antarctic measurement sites. In this paper, the time series of the noon UV index, noon total ozone, UVB, UVA, VIS and PAR measurements for the period March 2017 – May 2019 are presented in Figs. 6 and 7. Noon time UV index refers to the UV index measured at minimum SZA of the day. The maximum noon UV indices of the spring–autumn seasons were 6 and 9 in 2017–2018 and 2018–2019, respectively. In 2017–2018, the maximum noon UV index was measured both in the spring (October) and in the summer (January). In 2018–2019 the maximum was measured in the spring when the station was inside the polar vortex (Aun et al., 2019) and exceeded the summer noon maximum UV index which was 8. For the months October – February total ozone was lower in 2018–2019 than 2017–2018 for most of the days (Fig. 6). This resulted in higher UV indices and UVB levels in 2018–2019 compared to 2017–2018. The effect

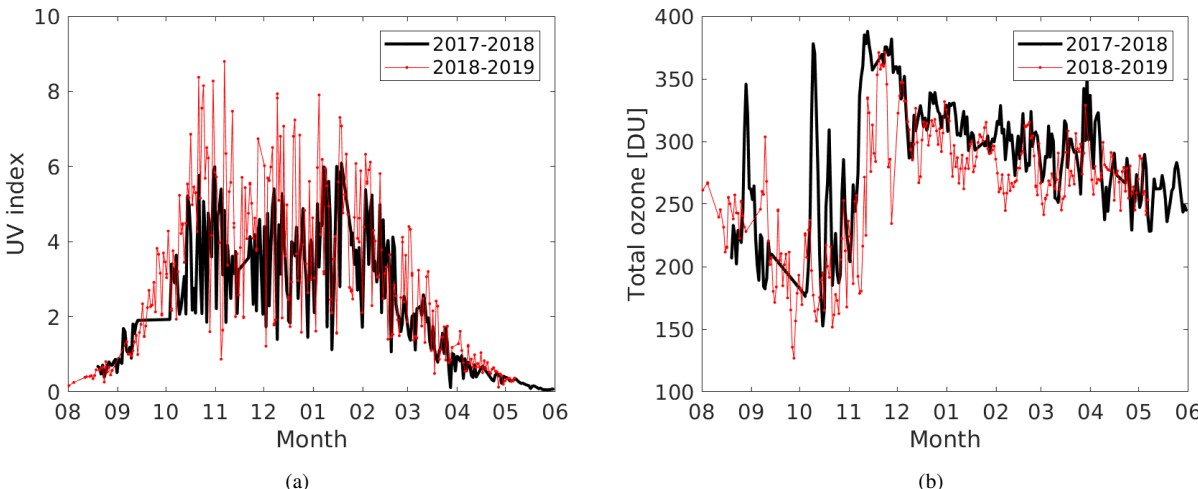

(a)  (b)

**Figure 6.** UV index (a) and total ozone (b) at noon calculated from the Marambio GUV radiometer measurements for the months August – May in 2017-2018 and 2018-2019.

of total ozone changes is more pronounced at shorter UVB wavelengths than at longer UVA wavelengths (Fig. 7), following the ozone absorption spectrum. Elevated UVB dose rates are detected in the spring and in the summer (Fig. 7c). The peak in October–November is due to the ozone hole, when the station is located below the stratospheric polar vortex. The reasons for the observed UV features are analyzed and discussed in more detail in Aun et al. (2019).

Variations in VIS and PAR are dominated by variations in cloudiness. Rapidly changing cloudiness is typical in Marambio (Aun et al., 2019), which is seen as high daily variation in all radiation measurements: UV, VIS and PAR. Surface albedo is higher during the spring months than during the summer months, which increases measured radiation in the spring at both shorter UV wavelengths and longer VIS and PAR wavelengths. This results in non-symmetric distribution of daily maxima; with higher spring time VIS radiation and PAR than in the autumn at similar SZAs. The time series of VIS and PAR radiation

suggest that there were more cloud-free periods around noon during mid-November – January in 2018–2019 than during the same months in 2017 – 2018, as peaks values where higher during the second year.

Table 4 summarizes the monthly means of the UVB and UVA daily doses, the UVB and UVA daily maximum dose rates, and the VIS and PAR daily maxima for September–March in 2017-2019.

**Table 4.** Monthly means of GUV radiometer products measured at Marambio during the period March 2017 – April 2019. Months with more than 15 measurement days are included. One standard deviation is showed in the parenthesis. DD=daily dose, DMDR = daily maximum dose rate, DM=daily maximum. VIS is irradinace measured at 555 nm.

| UVB DD [kJm$^{-2}$] | Jan | Feb | Mar | Sept | Oct | Nov | Dec |
|---|---|---|---|---|---|---|---|
| 2017 | NaN | NaN | 16.19( 7.06) | NaN | 33.55(11.03) | 41.38( 8.00) | 41.53(11.75) |
| 2018 | 42.62(14.72) | 34.60(11.19) | 14.09( 5.73) | 17.74( 7.92) | 44.04(12.55) | 44.89(14.28) | 50.56(15.25) |
| 2019 | 45.86(15.25) | 31.62(11.59) | 16.36( 9.14) | NaN | NaN | NaN | NaN |
| **UVA DD [MJm$^{-2}$]** | | | | | | | |
| 2017 | NaN | NaN | 0.50( 0.20) | NaN | 0.78( 0.25) | 1.18( 0.26) | 1.04( 0.31) |
| 2018 | 1.04( 0.37) | 0.91( 0.28) | 0.45( 0.14) | 0.45( 0.13) | 0.89( 0.22) | 1.04( 0.38) | 1.24( 0.38) |
| 2019 | 1.08( 0.36) | 0.79( 0.26) | 0.51( 0.24) | NaN | NaN | NaN | NaN |
| **UVB DMDR [$\mu$W cm$^{-2}$]** | | | | | | | |
| 2017 | NaN | NaN | 37.64(13.68) | NaN | 75.96(22.54) | 81.56(12.57) | 91.19(19.76) |
| 2018 | 92.91(24.68) | 77.46(20.57) | 33.40(13.61) | 43.24(18.21) | 101.67(27.97) | 103.80(29.60) | 108.02(24.54) |
| 2019 | 97.55(33.39) | 73.70(26.00) | 36.56(18.39) | NaN | NaN | NaN | NaN |
| **UVA DMDR [mW cm$^{-2}$]** | | | | | | | |
| 2017 | NaN | NaN | 2.39( 0.64) | NaN | 3.18( 0.69) | 4.32( 0.59) | 4.27( 0.92) |
| 2018 | 4.12( 1.17) | 3.98( 0.89) | 2.35( 0.66) | 2.19( 0.33) | 3.58( 0.78) | 4.38( 1.34) | 4.93( 1.09) |
| 2019 | 4.31( 1.42) | 3.60( 0.96) | 2.41( 0.82) | NaN | NaN | NaN | NaN |
| **VIS DM [$\mu$W cm$^{-2}$nm$^{-1}$]** | | | | | | | |
| 2017 | NaN | NaN | 75.61(18.85) | NaN | 96.16(22.23) | 132.00(21.03) | 132.10(34.04) |
| 2018 | 120.14(39.14) | 120.72(28.75) | 76.18(23.40) | 68.39(15.43) | 104.92(24.77) | 123.58(41.62) | 143.86(38.00) |
| 2019 | 123.36(47.74) | 103.05(33.12) | 72.73(26.88) | NaN | NaN | NaN | NaN |
| **PAR DM [$\mu$E cm$^{-2}$s$^{-1}$]** | | | | | | | |
| 2017 | NaN | NaN | 0.10( 0.03) | NaN | 0.13( 0.03) | 0.18( 0.03) | 0.18( 0.05) |
| 2018 | 0.16( 0.05) | 0.16( 0.04) | 0.10( 0.03) | 0.10( 0.01) | 0.14( 0.03) | 0.17( 0.06) | 0.19( 0.05) |
| 2019 | 0.16( 0.06) | 0.14( 0.04) | 0.10( 0.04) | NaN | NaN | NaN | NaN |

5    Figure 8 shows ratios of UV irradiances at 305, 313, 320 and 380 nm compared to VIS irradiance at 555 nm in 2017–2018 and 2018–2019. For wavelengths which are affected by ozone absorption (mainly 305 and 313 nm), the spring time low ozone episodes are clearly identified as increased ratios (October and November). In the autumn, the radiation path in the atmosphere becomes longer day by day, so that the attenuation of UV radiation is stronger than that of VIS radiation. This is seen at short

wavelengths as decrease of the UV/VIS ratio from summer to late autumn. The longer the atmospheric path is, the higher is the path-integrated total ozone concentration which can absorb UV radiation. The opposite is seen in the spring ratios. UV-A radiation, irradiances at 320 and 380 nm, is only partially absorbed by ozone, and the effect is smaller.

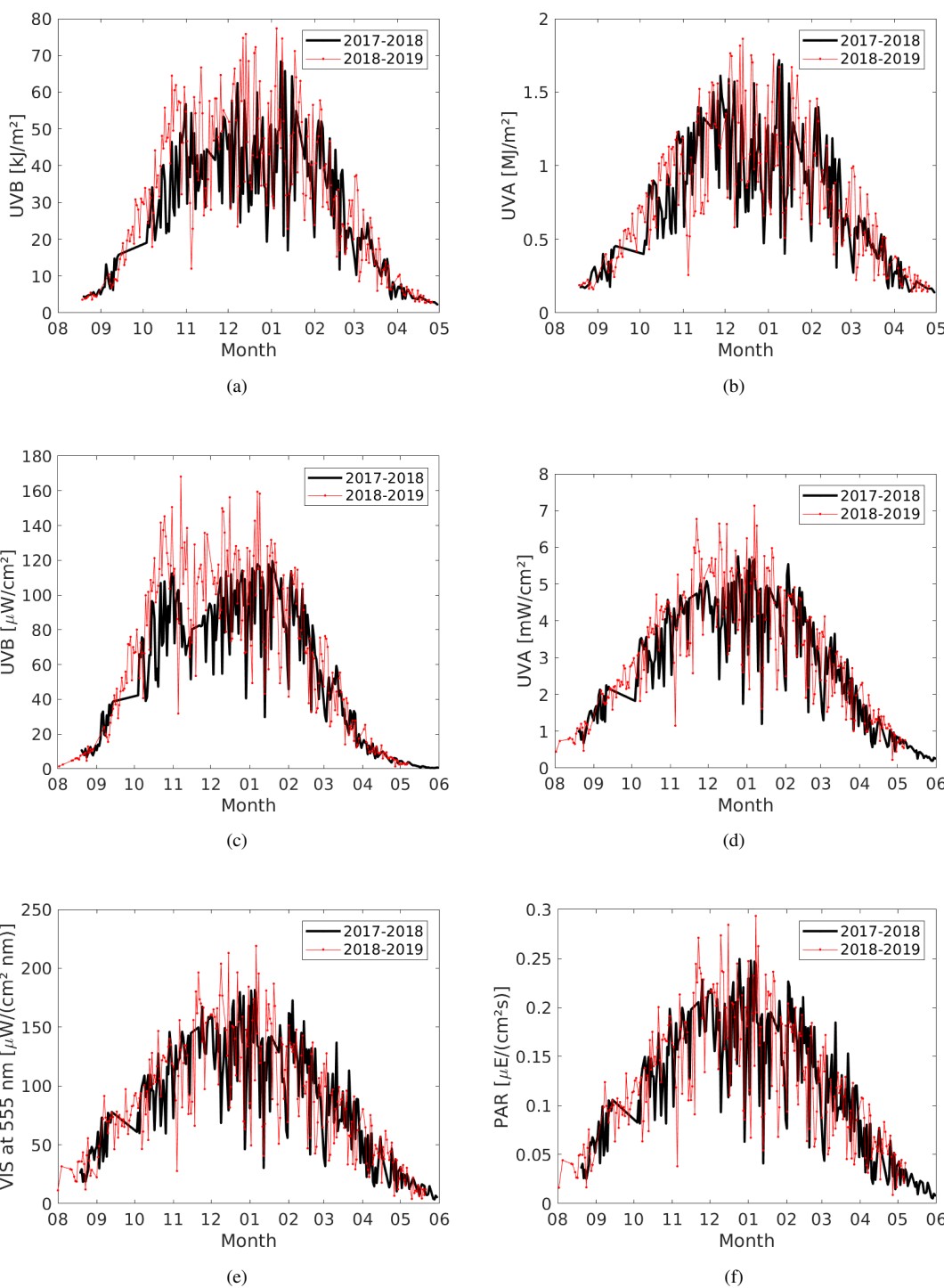

**Figure 7.** UV and visible radiation time series for the months August – May during periods 2017-2018 and 2018-2019. A) UVB daily dose, b) UVA daily dose, c)UVB daily maximum, d) UVA daily maximum, e) daily maximum VIS at 555 nm and f) PAR daily maximum.

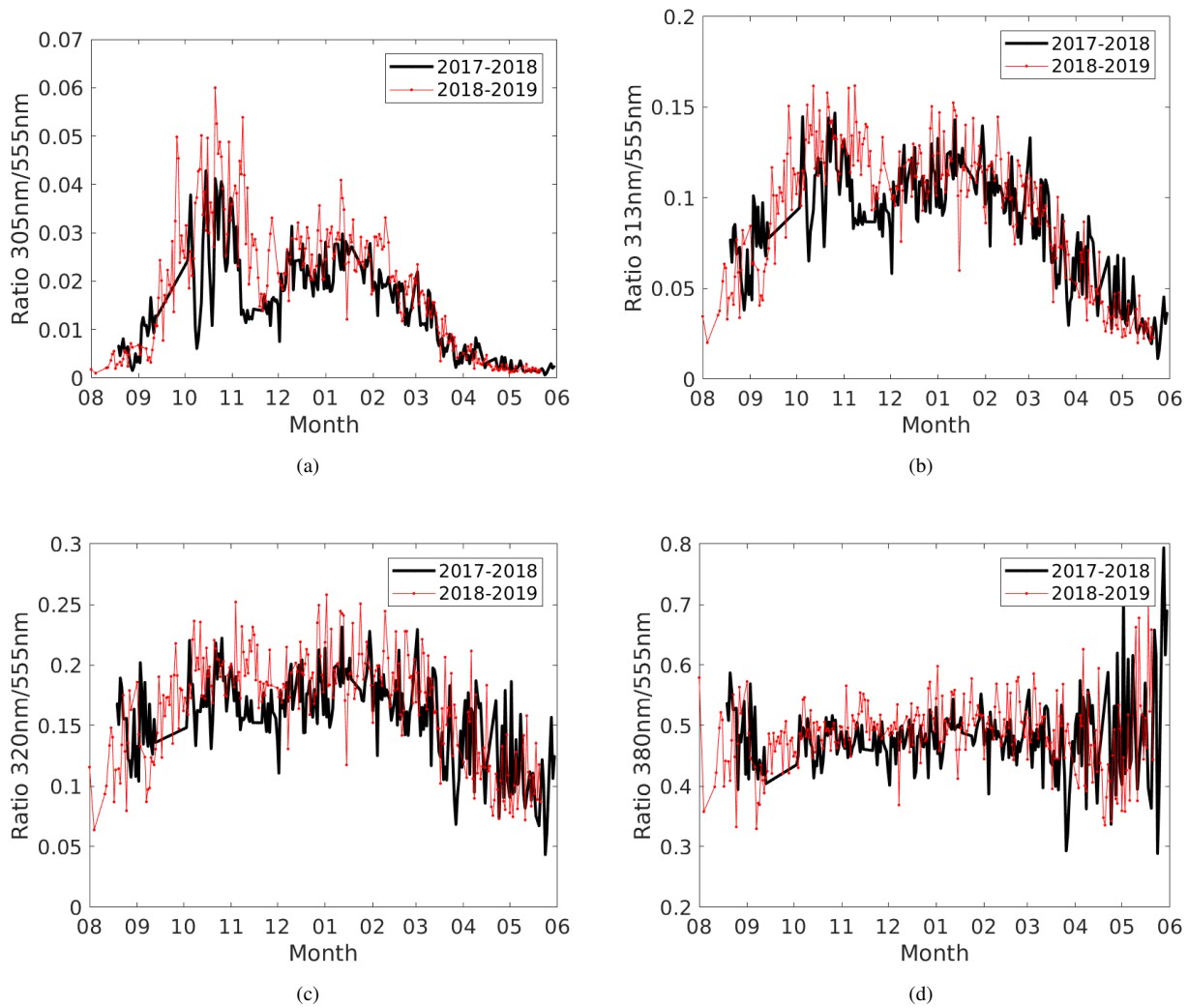

**Figure 8.** Ratio of daily maximum UV irradiances and VIS irradiances for the months August – May during periods 2017-2018 and 2018-2019. UV irradiances at A) 305 nm, b) 313 nm, c)320 nm and d) 380 nm.

## 5 Data availability

The studied datasets of daily maximum irradiances at five UV channels and one VIS channel, daily maximum UVB/UVA and PAR dose rates, noon UVB/UVA and PAR dose rates, noon total column ozone and UVB/UVA daily doses are freely available at Zenodo, http://doi.org/10.5281/zenodo.3688700 (Lakkala et al., 2019). The additional datasets including biologically
weighted dose rates and daily doses are available from the authors.

## 6 Discussion and Conclusions

Marambio's new GUV multifilter radiometer measurements cover the spectral range from the UV to the VIS, and thus can be used for assessing the spectral effects of changes in factors affecting the amount of solar UV and VIS radiation at the surface: aerosols, albedo, cloudiness and total ozone. Those factors are also influenced by climate change, and thus the measurements
serve as important tools for assessing effects of a changing climate on the Antarctic environment.

The recovery of the stratospheric ozone layer has started (WMO, 2018b) and the GUV total ozone measurements will be used to assess features of the recovery in Antarctica. The signs of the recovery can be detected by the UV channels of the radiometer, as actual measurements can be compared with the time series of the Antarctic-NILU-UV network and other UV measurements performed in Antarctica (Aun et al., 2019). The time series of the first two years of GUV measurements show
that the recovery is not seen as linear increase in UV levels, and clear differences exist between 2017–2018 and 2018–2019: the effect of the spring time ozone depletion was more pronounced in 2018–2019 than in 2017–2018 with maximum noon UV indices of 9 and 6, respectively. This is however far from the maximum UV index measured within the Antarctic NILU-UV network during the years 2000–2010, when a UV index of 12 was measured at Marambio in the spring (November) 2007 (Lakkala et al., 2018b). For comparison: at Arctic sites the spring time stratospheric ozone loss doesn't increase UV levels
to reach the yearly maxima, even during years with severe ozone depletion (Bernhard et al., 2013). In the Arctic, UV index maxima are observed in the summer. For example in Sodankylä, Finland, which is a site affected by stratospheric ozone loss, a maximum UV index of 6 has been measured in the summers of the years 2011 and 2013 (Lakkala et al., 2016).

The Antarctic UV measurements of Marambio can be directly compared to the Arctic UV measurements using the results from solar comparisons between Marambio's GUV radiometer, PMOD-WRC and FMI's spectroradiometers in Sodankylä. The
results show that the measurements agreed within ±6% for SZA < 60°. These yearly comparisons with Arctic measurements are possible because two GUVs have been purchased for Marambio's measurements: one is measuring at Marambio, while the other is participating in solar comparisons. In addition, regular absolute calibration against high-quality spectroradiometers is part of the quality assurance. The first re-calibration occurred in the spring of 2019. Both solar comparisons and regular absolute calibrations are crucial to obtain homogenized long-term time series (Johnsen et al., 2008), which can serve multidisciplinary
research communities. In addition, these high-quality measurements can be used for validation of satellite data, e.g., validation of UV products based on TROPOMI measurements on board the Sentinel-5 Precursor satellite (Lindfors et al., 2018). They are highly valuable, as the ground based measurement network is very sparse at high latitudes.

Ultraviolet radiation products calculated from Marambio's GUV measurements include UV dose rates and doses calculated using 10 different biological action spectra related to UV effects on, e.g., skin, plant, anchovy and phytoplankton. All these UV products can be used when assessing the effects of climate change on the whole Antarctic ecosystem. The effects are not self-evident even though the melting of ice sheet and snow (e.g., Shepherd et al., 2018) due to temperature increase (Steig et al., 2009) is well recognized. The uncertainty is mostly due to complicated atmosphere-sea-land feedback mechanisms, (e.g., Wang et al., 2019). In future, GUV's VIS radiation time series can be used together with ceilometer measurements and synoptic cloud observations to assess the impact of climate change on cloudiness. Together with the ongoing albedo and aerosol measurements in Marambio they form a complex measurement system for atmospheric radiation studies.

*Author contributions.* K. Lakkala: Primarily responsible for the QA of the UV data, analyzed the data and led the manuscript preparation.

M. Aun: Programmed GUV data processing for the Marambio station, participated in data analysis and contributed to the writing of the manuscript.

R. Sanchez: Responsible in SMN for Marambio UV measurements. Contributed to the writing of the manuscript.

G. Bernhard: Performed the absolute calibration of the GUV radiometers, programmed UV data processing program and contributed to the writing of the manuscript.

E. Asmi: Data analysis of Marambio cloudiness data. Contributed to the writing of the manuscript.

O. Meinander: Responsible for the UV albedo measurements at Marambio. Contributed to the writing of the manuscript.

F. Nollas: Data analysis of Marambio cloudiness data. Contributed to the writing of the manuscript.

G. Hülsen: Performed the QASUME site audit in Sodankylä in 2018. Processed QASUME reference UV data.

T. Karppinen: Sodankylä Brewer ozone processing and QC/QA. Contributed to the writing of the manuscript.

V. Aaltonen: Responsible for the PFR measurements at Marambio. Contributed to the writing of the manuscript.

A. Arola: Overseeing the work in the group and contributed to the writing of the manuscript.

G. de Leeuw: Group leader during the set up of Marambio UV measurements. Contributed to the writing of the manuscript.

*Competing interests.* No competing interests are present.

*Acknowledgements.* We thank the operators of the GUV radiometers in Marambio. The operators of the Finnish Brewers in Sodankylä are acknowledged for the daily operation and QC of the measurements. Edith Rodriguez is acknowledged for help with logistics. Hanne Suokanerva and Riika Ylitalo are acknowledged for data dissemination. Outi Meinander was supported by the Nordic Center of Excellence CRAICC, Academy of Finland Center of Excellence program (project no. 272041), Ministry for Foreign Affairs of Finland IBA-project (no. PC0TQ4BT-25), EU-Interact-BLACK-project (H2020 Grant Agreement No. 730938) and the Academy of Finland NABCEA-project (no. 296302).

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
