# Peer review of "New continuous total ozone, UV, VIS and PAR measurements at Marambio 64°S, Antarctica"

_Earth System Science Data, 2019_

## Referee Comment (RC1) · Anonymous Referee #1 · 19 Dec 2019

General comments:

The manuscript by Lakkala et al. discusses a new dataset which consists of UV and visible solar irradiance measurements, effective biological doses, and the total column of ozone. The measurements of the solar irradiance – from which the effective doses and the total column of ozone have been calculated – are performed at the Antarctic station of Marambio. Part of the discussion has been focused on the procedures which ensure the good quality of the measurements. The described dataset is of high scientific significance since analysis of the products would contribute to the assessment of the impacts of changes in total ozone and climate over the sensitive environment of Antarctica. The manuscript is within the scope of the journal and should be published after minor revision by the authors.

[Figure]

What I mainly miss is some discussion (maybe a small paragraph) regarding the uncertainty in the GUV measurements performed at Marambio. Comparison with other instruments provides very strong evidence of the reliability of the measurements. Are however the calculated differences representative for the overall measurement uncertainties? I believe that some discussion regarding the magnitude of the overall uncertainties – not necessarily a precise determination of the uncertainty budget - and the main uncertainty factors would be useful for the readers, as well as for people interested for the data.

More analytical comments are provided below:

References are missing at several points in the introduction. I suggest adding references to the following phrases: P2, L4 – 5: "The international . . . UV levels" P2, L5 – 6: "The . . . materials" P2, L17 – 19: "As . . . effects" P2, L26 – 27: "Both . . . Ocean" P2, L27 – 28: "Changes . . . series"

P3, L6: Please replace "measurements were" with "was"

P7, L7: in "ki", i is an index

P7, L8 – 9: "A sensitivity . . . time" is there any reference which can be used to support this statement?

P8, L7: Was the sky clear in August 2016? Please specify since the conclusions from Figure 2 might be slightly different if the sky was cloudy.

Figure 2: Could the apparent dependence of the ratio on SZA (or part of it) be a masked effect of temperature on the response of any of the instruments?

Section 3.3: Is this change in the response of the instrument (taking place within the 1 or 2 years between sequential calibrations) somehow taken into account, e.g. by interpolating the calibration factors?
* * *
[Figure]

2019.

---

## Referee Comment (RC2) · Anonymous Referee #2 · 13 Jan 2020

General comments: The manuscript presents and discusses a dataset for the Antarctic station Marambio, that is based on GUV multiband filter instruments installed in 2017, providing a continuation of the 2000-2010 NILU-UV network series at Marambio. The time series are accessible at https://zenodo.org/record/3553634#.XhsiBchKhaQ, and contain several data products that are essential for satellite ground truthing and assessing the impact of global climate change in the Antarctic region. The paper also present infrastructure and data resources available for scientific use at Marambio, facilitating scientific co-ordination in Antarctica. The monitoring program conducts strict quality site control and quality assurance, including repeatably intercomparisons with spectroradiometers, and by using one of the GUV instruments as travelling reference for the other.

[Figure]

Unfortunately, after analyzing the data sets and presentations in the manuscript, I found several flaws regarding use of measurement units, as well as suspecting anomalous results in final data. The manuscript is within the scope of the journal, but the data sets should be re-evaluated, and manuscript revised before publication.

Specific comments. Page 1 in Abstract: lines 9-11: A long list of final data products is stated being available – however in the internet link provided I found only a subset available at the given repository (e.g. 10 biologially effective dose rates and corresponding doses are missing). Please, be more specific. A screendump of files available (attached figure).

Page 1 line 14-15 in Abstract: "Average daily maximum UVB dose rates 7.6 – 10.2 kW/mˆ2". Compare these numbers for the UVB with the Sun's total integrated solar irradiance at the top of the atmosphere – 1.366 kW/mˆ2 (https://wattsupwiththat.com/2018/09/19/how-constant-is-the-solar-constant/). One might suspect a misprint, that the prefix k (kilo) should be omitted, but even in that case, 7.6-10.2 W/mˆ2 is almost a factor 10 above realistic natural surface irradiance levels, compared with quality controlled measurement data for an equivalent latitude and network station (e.g. mountain station Finse in the UV-monitoring network in Norway, spanning latitudes 58-78 N https://github.com/uvnrpa). The same applies to Table 4, UVB DMDR [kW/mˆ2] and UVA DMDR [kW/mˆ2], as well as Figure 5C and 5D, where units and numbers also look anomalous.

Page 3-4, section 2.1 Marambio station and section 2.2: The text relates much to the same content provided in section 2.1.1 and section 2.2.1 in another publication under discussions by the two first authors, where the same datasets are applied https://www.atmos-chem-phys-discuss.net/acp-2019-896/acp-2019-896.pdf Figure 4A and Figure 4B on page 10 in Lakkala et al. is almost identical with Figure 5 and Figure 6 in the second paper submitted by Aun et al. 2019. Although the two papers deal with different topics: one presenting the data sets, QA/QC and resources, and the second paper an analysis of UVI and erythemal doses in relation to the long-term series, there

are redundancies between the two papers.

Page 5, Figure 1c and Figure 1 d: Units are given as uW/mˆ2. The units should probably be given as uW/cmˆ2 (difference 10ˆ4).

Pages 7-8. The section describes calibrations and comparisons of UVI against spectroradiometers SUV, Brewers and QASUME. I miss a comparison which includes other data products as well in order get an estimate of overall uncertainties.

Table 5 and Table 6. Please, consider if this information is too detailed in this context.

Page 14, Figure 5E: Units is given as mW/cmˆ2/nm at 555 nm. Realistic units is uW/cmˆ2/nm (difference 10ˆ3). Page 14, Figure 5F: PAR, units given as E/mˆ2/s. Realistic units is E/cmˆ2/s (difference 10ˆ4).

Page 16, Section 5 Data availability. The units of most datasets look correct. However, the six data files max305nm.dat etc are expressed in units W/cmˆ2/nm. The units should probably be uW/cmˆ2/nm (difference 10ˆ6). I have plotted the spectral irradiance data from these files, and changed observations to appropriate units, and made a model comparison, see attached figure. The model takes as input the total ozone given in file GUV_UVB_UVA_PAR_O3_noon.dat, the noon SZA for Marambio, assuming snowfree ground (albedo given as 0.2 to 0.3 for November/December in Aun et al. 2019) and assuming clear sky conditions.

Irradiance values of observations look anomalous for the 313 and 320 nm (factor almost 2x), but reasonable for 305 nm, 340 nm and 380 nm. You can see this by observing the differences in spectral irradiances for increasing wavelengths of observations and modelled data: Subset of data covering 2017/2018, wavelengths 305-313-320-340-380 nm: Observations: 4-26-16-54-74 uW/cmˆ2/nm Model predications: 4-15-26-54-64 uW/cmˆ2/nm (continuous increase, matching observations at the three wavelengths 305, 340 and 380 nm). Page 16, line 26-27. "..crucial to obtain homogenized long-term series..". Here I miss references to the international intercomparison of multiband filter radiometers, held in Oslo 2005: GAW report no. 179 / WMO/TD-No. 1454. Geneve: World Meteorological Organization. 2008.

Caption to figure DailyMax_Marambio.png Figure – a compilation of daily max spectral irradiance at 5 wavelengths provided by the authors for the Antarctic station Marambio (where I have rescaled data in realistic units (uW/cmˆ2/nm) – versus clear sky values derived from a radiative transfer model (libradtran), with input total ozone amounts based on the authors final results.

[Figure]

**Fig. 1.**

[Figure]

Fig. 2.

---

## Author Comment (AC1) · 26 Feb 2020

**Final Author comments**

Authors' response to Referee #1 and Referee #2 comments on " New continuous total ozone, UV, VIS and PAR measurements at Marambio 64°S, Antarctica" by Kaisa Lakkala et al.

The authors thank the Referees for constructive comments and reply to all comments here below. The answer is structured as follow: (1) comments from Referee, (2) author's response, (3) author's changes in the manuscript.

**Referee #1**

(1) What I mainly miss is some discussion (maybe a small paragraph) regarding the uncertainty in the GUV measurements performed at Marambio. Comparison with other instruments provides very strong evidence of the reliability of the measurements. Are however the calculated differences representative for the overall measurement uncertainties? I believe that some discussion regarding the magnitude of the overall uncertainties – not necessarily a precise determination of the uncertainty budget - and the main uncertainty factors would be useful for the readers, as well as for people interested for the data.

(2) A new paragraph (see below) has been added to Section 3.1. discussing the uncertainties of GUV data products (e.g., UV index, UVB, UVA, DNA damage).

(3) "The uncertainty of GUV data products is composed of (i) the uncertainty of SUV-100 measurements, (ii) the uncertainty of the transfer of the calibration from the SUV-100 to the GUV, (iii) the uncertainty of the conversion from response-weighted irradiance to data products $D$, and (iv) the drift of GUV calibrations. The uncertainty of SUV-100 measurements has been assessed by Bernhard et al. (2004). The expanded uncertainty (coverage factor $k = 2$, corresponding to a level of confidence of approximately 95 %) of the UV index and DNA-damaging irradiance varies between 5.8 and 6.4 %. The upper limit of errors in the SUV-to-GUV calibration transfer was estimated to ±2% from the reproducibility of the vicarious calibration method. The uncertainty in calculating the UV index from response-weighted irradiance using Eq. (1) was assessed by Dahlback (1996). For SZAs between 0 and 80° and total ozone between 200 and 500 DU, the approximations implied in using Eq. (1) agreed to within ±5% with exact radiative transfer calculations. However, larger errors were found for total ozone columns smaller than 200 DU or for SZAs larger than 80° (when absolute values of the UV Index are small). Bernhard et al. (2005) compared measurements of UV-B irradiance, UV-A irradiance, the UV index, and DNA-damaging performed at the South Pole with a SUV-100 and a GUV radiometer, which was calibrated with the method described in this paper. For SZAs smaller than 80°, measurements by the two instruments agreed on average to within ±2.5%, and the standard deviation of the ratio of GUV/SUV data was smaller than 4.0% for the four data products. The magnitude of these variations is in good agreement with the theoretical calculations by Dahlback (1996). Lastly, the uncertainty attributed to drifts was calculated from the observed change of 5% in the responsivity of the GUV's 305 nm channel. The four uncertainty components were combined in quadrature and multiplied with a coverage factor of $k = 2$. For SZAs smaller than 80°, the expanded ($k = 2$) uncertainty is 9% for UV-B irradiance, the UV index, and DNA-damaging irradiance."

New reference added: Bernhard, G., Booth, C. R., and Ehramjian, J. C.: Version 2 data of the National Science Foundation's Ultraviolet Radiation Monitoring Network – South Pole, J. Geophys. Res., 109, D21207, https://doi.org/10.1029/2004JD004937, 2004.

Specific comments:

(1) References are missing at several points in the introduction. I suggest adding references to the following phrases: P2, L4 – 5: "The international . . . UV levels" P2, L5 –6: "The . . . materials" P2, L17 – 19: "As . . . effects" P2, L26 – 27: "Both . . . Ocean" P2, L27 – 28: "Changes . . . series"
(2) References have been added.
(3) References have been added: "The international Montreal Protocol was signed in 1987 to restrict the use of ozone depleting substances and thus protect the whole ecosystem from excessive UV levels (e.g., Barnes et al. (2019); WMO (2018b)). The shortest UV wavelengths are known to cause skin cancer, sunburn and cataract, and they can also harm plants, animals and materials (e.g., EEAP (2019)).

As the stratospheric ozone depletion has influenced both stratospheric and surface climate, e.g., cooling of Antarctic stratosphere and southward shift of mid-latitude rain, and the Southern Ocean temperature and circulation, the recovery is expected to have the opposite effects (WMO, 2018a).

Both UV and PAR affect micro-organisms living in Antarctic ice and the Southern Ocean (Deppeler and Davidson, 2017; Häder et al., 2014).

Changes in the amount of aerosols and pollution as well as changes in sea ice extension or ground albedo are also reflected in both UV and VIS radiation time series (Fountoulakis et al., 2014; Wild, 2009)."

(1) P3, L6: Please replace "measurements were" with "was"
(2) Replaced as suggested.
(3) Replaced as suggested.

(1) P7, L7: in "ki", i is an index
(2) We agree.
(3) Manuscript updated following the comment.

(1) P7, L8 – 9: "A sensitivity . . . time" is there any reference which can be used to support this statement?
(2) The statement was based on unpublished data, which at present cannot be supported with a reference. We deleted this sentence.
(3) Sentence was deleted.

(1) P8, L7: Was the sky clear in August 2016? Please specify since the conclusions from Figure 2 might be slightly different if the sky was cloudy.
(2) Yes, there was clear sky on 24 August 2016.
(3) The information was added to the manuscript and the following sentence of Section 3.2 was updated: "For clear sky and SZAs smaller than 60°, UV indices measured by both GUV radiometers agreed within 1% of each other and the Brewer spectroradiometer."

(1) Figure 2: Could the apparent dependence of the ratio on SZA (or part of it) be a masked effect of temperature on the response of any of the instruments?
(2) The internal temperature of the GUV radiometer was monitored and kept constant at 40ºC, which ensured that no drift of response was due to temperature change. The UV data of FMI's spectroradiometer was temperature corrected using the method presented in Lakkala et al. 2008. The reference spectroradiometer QASUME was temperature stabilized during the measurement campaign. The small dependence of the ratio on SZA is likely due to a combination of serval factors, including difference in the angular response of the instruments (including incomplete corrections of cosine

errors), the approximation in calculating the UV index with Eq. (1) and small time shifts between GUV and spectroradiometric measurements as the recording of a spectrum takes several minutes. (3) The information of Brewer temperature correction and temperature stabilization of the QASUME instrument was added to the text in section 3.2. Referring to the Brewer: " The measurements are cosine and temperature error corrected..." and to the QASUME "The measurements are temperature stabilized...".

Reference: Lakkala, K., Arola, A., Heikkilä, A., Kaurola, J., Koskela, T., Kyrö, E., Lindfors, A., Meinander, O., Tanskanen, A., Gröbner, J., and Hülsen, G.: Quality assurance of the Brewer spectral UV measurements in Finland, Atmos. Chem. Phys., 8, 3369–3383, https://doi.org/10.5194/acp-8-3369-2008, 2008.

(1) Section 3.3: Is this change in the response of the instrument (taking place within the 1 or 2 years between sequential calibrations) somehow taken into account, e.g. by interpolating the calibration factors?
(2) The change in the response of the instrument is not taken into account. The manuscript has been updated including the information.
(3) The manuscript has been updated including the information. The following sentence was added to section 3.3.: "The drift was considered to be within the uncertainties of the measurements and the time series was not corrected for. "

**Referee #2**
(1) Unfortunately, after analyzing the data sets and presentations in the manuscript, I found several flaws regarding use of measurement units, as well as suspecting anomalous results in final data. The manuscript is within the scope of the journal, but the data sets should be re-evaluated, and manuscript revised before publication.
(2) The authors thank the Referee for the careful review and agree with the problem in the units of the data. The mistake was found to be in the data-analyse phase, not in the original data set.
(3) The units and data have been corrected and the updated dataset is uploaded to http://doi.org/10.5281/zenodo.3688700.

Specific comments:

(1) Page 1 in Abstract: lines 9-11: A long list of final data products is stated being available – however in the internet link provided I found only a subset available at the given repository (e.g. 10 biologially effective dose rates and corresponding doses are missing). Please, be more specific. A screendump of files available (attached figure).
(2) We included in the available dataset the studied data sets, which are irradiances at five UV channels and one VIS channel, daily maximum UVB/UVA and PAR dose rates, noon UVB/UVA and PAR dose rates, noon total column ozone and UVB/UVA daily doses. The data including weighting using 10 different biologically active were not studied in this specific paper, even if it is routinely derived from the actual measurements. That data is available from the authors.
(3) The Abstract has been updated to be more specific: "The studied dataset, including daily maximum irradiances at five UV channels and one VIS channel, daily maximum UVB/UVA and PAR dose rates, noon UVB/UVA and PAR dose rates, noon total column ozone and UVB/UVA daily doses, is freely accessible at http://doi.org/10.5281/zenodo.3688700. (Lakkala et al., 2019)."

The Data availability -section has been updated to be more specific: "The studied datasets of daily maximum irradiances at five UV channels and one VIS channel, daily maximum UVB/UVA and PAR dose rates, noon UVB/UVA and PAR dose rates, noon total column ozone and UVB/UVA daily doses are freely available at Zenodo, http://doi.org/10.5281/zenodo.3688700. (Lakkala et al., 2019). The additional datasets including biologically weighted dose rates and daily doses are available from the authors."

(1) Page 1 line 14-15 in Abstract: "Average daily maximum UVB dose rates 7.6 – 10.2 kW/m^2". Compare these numbers for the UVB with the Sun's to- tal integrated solar irradiance at the top of the atmosphere – 1.366 kW/m^2 (https://wattsupwiththat.com/2018/09/19/how-constant-is-the-solar-constant/). One might suspect a misprint, that the prefix k (kilo) should be omitted, but even in that case, 7.6-10.2 W/m^2 is almost a factor 10 above realistic natural surface irradiance levels, compared with quality controlled measurement data for an equivalent latitude and network station (e.g. mountain station Finse in the UV-monitoring network in Nor- way, spanning latitudes 58-78 N https://github.com/uvnrpa). The same applies to Table 4, UVB DMDR [kW/m^2] and UVA DMDR [kW/m^2], as well as Figure 5C and 5D, where units and numbers also look anomalous.

(2) The authors thank the Referee for the careful review and agree with the problem in the units of the data. The mistake was found to be in the data-analyse phase, not in the original data set. The units and data have been corrected and the updated dataset is uploaded in http://doi.org/10.5281/zenodo.3688700.

(3) The units have been corrected and the Table 4 and Figure 5 (now 7) have been updated. The sentence in the abstract has been updated to be " Average daily maximum UVB dose rates 76 – 102 uW/cm^2..."

(1) Page 3-4, section 2.1 Marambio station and section 2.2: The text relates much to the same content provided in section 2.1.1 and section 2.2.1 in another publication under discussions by the two first authors, where the same datasets are applied https://www.atmos-chem-phys-discuss.net/acp-2019-896/acp-2019-896.pdf . Figure 4A and Figure 4B on page 10 in Lakkala et al. is almost identical with Figure 5 and Figure 6 in the second paper submitted by Aun et al. 2019. Although the two papers deal with different topics: one presenting the data sets, QA/QC and resources, and the second paper an analysis of UVI and erythemal doses in relation to the long-term series, there are redundancies between the two papers.

(2) As commented by the Referee, the two papers have different topics. This one, Lakkala et al., focuses on the set up of the new measurement system, quality assurance and it's data set, and gives example of the use of the data by presenting products (UV, PAR and VIS) measured during the first two years of operation. Aun et al., 2019 analyse the characteristics of UV index and erythemally weighted UV time series during the first two years, compare them to earlier measurements (2000-2008) and to measurements from other Antarctic sites.

The text of sections 2.1. and 2.2. has been revised to avoid same content/phrasing. Naturally the sections include in some extent similar content than the sections 2.1.1 and 2.2.1. in Aun et al., as the sections are about the same measurements. The revised version of the section 2.2.1. of Aun et al., 2019 has been updated and includes now references to Lakkala et al.

We agree that the Figures 4A and 4B in Lakkala et al. are very similar to Figures 5 and 6 of Aun et al. However, we think that for a reader, the UV index and total column ozone are the most interesting and familiar, and we would prefer to keep the Figures in our manuscript. To avoid overlapping, we have not plotted the same quantities: In Aun et al. the plotted data in Figure 5 is **daily maximum** UV index and in Figure 6 **daily average** total ozone, while in Lakkala et al. the Figure 4a and b includes UV index

and total ozone measured **at noon**. At Marambio, total column ozone can change during the day so that the daily average is not the same as the one measured at noon. Regarding UV index, the noon value is the maximum for a clear sky day, but as Marambio has frequent changing cloudiness conditions, the daily maximum UV index can be measured either before or after noon, if the sky is cloudy during noon, but free from clouds later or earlier during the day.

The manuscript includes already the following sentences to clarify the differences between the two papers (First two sentences in Section 4.): "The erythemally weighted UV, maximum UV index and daily average total ozone time series were discussed in Aun et al.(2019) and the results were compared to measurements from other Antarctic measurement sites. In this paper, the time series of noon UV index, noon total ozone, UVB, UVA, VIS and PAR measurements for the period March 2017 – May 2019 are presented in Figs. 4 and 5 (Lakkala et al., 2019)."

(3) The text of sections 2.1. and 2.2. has been revised to avoid same phrasing.

(1) Page 5, Figure 1c and Figure 1 d: Units are given as uW/m^2. The units should probably be given as uW/cm^2 (difference 10^4).

(2) The authors agree.

(3) The Figures have been updated.

(1) Pages 7-8. The section describes calibrations and comparisons of UVI against spec-troradiometers SUV, Brewers and QASUME. I miss a comparison which includes other data products as well in order get an estimate of overall uncertainties.

(2) The comparison of UVB and total ozone have been added to the analysis. Comparisons of UVA, PAR and VIS products were not possible, as the spectroradiometer at Sodankylä doesn't measure these wavelengths. Bernhard et al., (2005) have validated several GUV radiometer products against a SUV spectroradiometer. The same methodology was used for the calibration of FMI's GUV radiometers, and the accuracy of the measurements can be assumed to be similar. A discussion about overall uncertainties has been added to the text in Section 3.1.

(3) Figures of UVB and total ozone column comparisons in Sodankylä have been added to the manuscript sections 3.2. and 3.2.1. The following discussion about the overall uncertainties has been added to the Section 3.1. :"The uncertainty of GUV data products is composed of (i) the uncertainty of SUV-100 measurements, (ii) the uncertainty of the transfer of the calibration from the SUV-100 to the GUV, (iii) the uncertainty of the conversion from response-weighted irradiance to data products $D$, and (iv) the drift of GUV calibrations. The uncertainty of SUV-100 measurements has been assessed by Bernhard et al. (2004). The expanded uncertainty (coverage factor $k = 2$, corresponding to a level of confidence of approximately 95 %) of the UV index and DNA-damaging irradiance varies between 5.8 and 6.4 %. The upper limit of errors in the SUV-to-GUV calibration transfer was estimated to ±2% from the reproducibility of the vicarious calibration method. The uncertainty in calculating the UV index from response-weighted irradiance using Eq. (1) was assessed by Dahlback (1996). For SZAs between 0 and 80° and total ozone between 200 and 500 DU, the approximations implied in using Eq. (1) agreed to within ±5% with exact radiative transfer calculations. However, larger errors were found for total ozone columns smaller than 200 DU or for SZAs larger than 80° (when absolute values of the UV Index are small). Bernhard et al. (2005) compared measurements of UV-B irradiance, UV-A irradiance, the UV index, and DNA-damaging performed at the South Pole with a SUV-100 and a GUV radiometer, which was calibrated with the method described in this paper. For SZAs smaller than 80°, measurements by the two instruments agreed on average to within ±2.5%, and the standard deviation of the ratio of GUV/SUV data was smaller than 4.0% for the four data products. The magnitude of these variations is in good agreement with the theoretical calculations by Dahlback (1996). Lastly, the

uncertainty attributed to drifts was calculated from the observed change of 5% in the responsivity of the GUV's 305 nm channel. The four uncertainty components were combined in quadrature and multiplied with a coverage factor of $k = 2$. For SZAs smaller than 80°, the expanded ($k = 2$) uncertainty is 9% for UV-B irradiance, the UV index, and DNA-damaging irradiance."

New reference added: Bernhard, G., Booth, C. R., and Ehramjian, J. C.: Version 2 data of the National Science Foundation's Ultraviolet Radiation Monitoring Network – South Pole, J. Geophys. Res., 109, D21207, https://doi.org/10.1029/2004JD004937, 2004.

(1) Table 5 and Table 6. Please, consider if this information is too detailed in this context.
(2) We agree and the tables were removed.
(3) The tables were removed.

(1) Page 14, Figure 5E: Units is given as mW/cm^2/nm at 555 nm. Realistic units is uW/cm^2/nm (difference 10^3). Page 14, Figure 5F: PAR, units given as E/m^2/s. Realistic units is E/cm^2/s (difference 10^4).
(2) We agree and the units have been updated. Note that the PAR is uE/cm^2/s.
(3) Units have been updated.

(1) Page 16, Section 5 Data availability. The units of most datasets look correct. However, the six data files max305nm.dat etc are expressed in units W/cm^2/nm. The units should probably be uW/cm^2/nm (difference 10^6). I have plotted the spectral irradiance data from these files, and changed observations to appropriate units, and made a model comparison, see attached figure. The model takes as input the total ozone given in file GUV_UVB_UVA_PAR_O3_noon.dat, the noon SZA for Marambio, assuming snowfree ground (albedo given as 0.2 to 0.3 for November/December in Aun et al. 2019) and assuming clear sky conditions.
(2) We agree with the unit problem. They have now been updated.
(3) Units have been updated to uW/cm^2/nm.

(1) Irradiance values of observations look anomalous for the 313 and 320 nm (factor almost 2x), but reasonable for 305 nm, 340 nm and 380 nm. You can see this by observing the differences in spectral irradiances for increasing wavelengths of observations and modelled data: Subset of data covering 2017/2018, wavelengths 305-313-320-340-380 nm: Observations: 4-26-16-54-74 uW/cm^2/nm Model predications: 4-15-26-54-64 uW/cm^2/nm (continuous increase, matching observations at the three wavelengths 305, 340 and 380 nm).
(2) We thank the Referee for the careful review of manuscript and for the model calculations. By downloading the data set from http://doi.org/10.5281/zenodo.3553634 and plotting it, we didn't get the same features than the Referee. We think that the irradiances of 320 and 313 nm have accidentally been mixed in the analysis of the Referee. Please find here below (Fig.1) a plot of irradiances at 305, 313, 320, 340 and 380 nm. We think that these time series match well with the model calculations presented by the author.

(3) None.

(1) Page 16, line 26-27. "..crucial to obtain homogenized long-term series..". Here I miss references to the international intercomparison of multi-band filter radiometers, held in Oslo 2005: GAW report no. 179 / WMO/TD-No. 1454.

Geneve: World Meteorological Organization. 2008.
(2) We added the reference.
(3) Reference added.

[Figure]

Figure 1. GUV irradiances measured at Marambio, plotted from the dataset 10.5281/zenodo.3688700.